# The Role of Preoperative Platelet-to-Lymphocyte Ratio as a Predictor for Incisional Hernias after Hand-Assisted Laparoscopic Liver Surgery for Metastatic Colorectal Cancer

**DOI:** 10.3390/jpm12030492

**Published:** 2022-03-18

**Authors:** Ahmad Mahamid, Omar Abu-Zaydeh, Muneer Sawaied, Natalia Goldberg, Riad Haddad

**Affiliations:** 1Department of Surgery, Carmel Medical Center, Haifa 3436212, Israel; mahamidam@yahoo.com (A.M.); oabuzaydeh@gmail.com (O.A.-Z.); moneerswaed@gmail.com (M.S.); 2The Ruth and Bruce Rappaport Faculty of Medicine, Technion, Israel Institute of Technology, Haifa 3525433, Israel; natalia.goldberg@gmail.com; 3Department of Radiology, Carmel Medical Center, Haifa 3436212, Israel

**Keywords:** liver surgery, hand-assisted laparoscopic surgery, incisional hernia, liver resection, colorectal liver metastasis

## Abstract

(1) Background: Hand-assisted laparoscopic surgery for liver resection is a globally established technique. In this study, we report on the incidence and risk factors for postoperative incisional hernia (IH) after hand-assisted laparoscopic surgery for colorectal liver metastasis. (2) Methods: This was retrospective analysis of 89 consecutive hand-assisted laparoscopic surgery for colorectal liver metastasis. (3) Results: Participants were 39 females and 50 males. Median age was 65 years, and in 63%, the BMI was ≥25. Postoperative complications were encountered in 18% of the patients. Seven patients (7.8%) had postoperative incisional hernia in the hand port site. There was significantly higher incidence of incisional hernia in overweight patients (BMI ≥ 25) (*p* = 0.04), and in cases with simultaneous liver and colon resection (*p* = 0.02). In univariant and multivariant analyses, simultaneous liver and colon resection (*p* = 0.004 and 0.03, respectively), and platelet-to-lymphocyte ratio ≤ 200 (*p* = 0.03, 0.04, respectively) were both independent risk factors for developing postoperative incisional hernia. (4) Conclusions: Both simultaneous liver and colon resection, and platelet-to-lymphocyte ratio ≤ 200 are independent risk factors for postoperative incisional hernia after hand-assisted laparoscopic surgery for colorectal liver metastasis.

## 1. Introduction

Since the first publications of laparoscopic nonanatomic liver resection and left lateral resection in the 1990s, laparoscopic liver resection (LLR) has been a widespread procedure, and it was expanded to segmentectomy, sectionectomy, hemihepatectomy, and the resection of posterosuperior segments [1,2,3].

Hand-assisted laparoscopic surgery (HALS) is a hybrid approach allowing for the surgeon to insert their hand via a port while maintaining pneumoperitoneum to carry out a laparoscopic procedure. This allows for tactile feedback and assistance with retraction, dissection, and hemostasis. The same incision is then used for specimen extraction [4]. The three consensus guidelines of Southampton, Louisville, and Morioka in laparoscopic liver resections estimate that pure laparoscopic liver resection, HALS, and the hybrid technique are equivalent and just a matter of surgeon preference. They also established the feasibility, safety, and long-term outcomes of LLR in comparison with the open approach [3,5,6].

In all types of minimal invasive procedures, there is still a need for abdominal wall incision to extract the resected specimen. The placement and length of incisions are variable and depend on surgeon preference with respect to the specific surgical technique and the specimen size. In HALS for liver resection, the length and place of the incision are predefined, usually in the range of 6–8 cm length in the upper right quadrant of the abdomen or midline. 

There are few data in the English literature that report on abdominal wall complications after HALS for liver resections. This study is the largest cohort that reports on incidence and risk factors for postoperative incisional hernia (IH) after HALS for colorectal liver metastasis.

## 2. Materials and Methods

All patients who underwent hand-assisted laparoscopic liver surgery for metastatic colorectal cancer between 2009 and December 2019 in Carmel Medical Center were identified from prospective maintained surgical databases. The study was conducted in accordance with the Declaration of Helsinki and was approved by the institutional review board (IRB) of Carmel Medical Center. Clinical data, including patient demographics, imaging work-up, perioperative treatment, surgical procedure, pathology results, and surgical–oncologic follow-up were retrospectively collected from electronic health records. 

All procedures were performed by the same surgical team under the direction of the same attending surgeon (R.H.). The HALS approach for CRLM was performed in the same way as described by Sadot et al. [7]. In brief, patients were put in a supine position with split legs. Three trocars (two 12 mm and one 5 mm) were inserted into the upper midline abdomen, and a hand-assisted device (GelPort, Applied Medical, Rancho Santa Margarita, CA, USA) was placed in the right abdomen. A right abdominal horizontal incision (7–8 cm) was performed. The surgeon dissected the adhesions, and then inserted the hand port and a 12 mm supraumbilical trocar into an adhesion-free area. We placed the GEL port and the trocars on the same portion of the abdomen for all patients, unrelated to the tumor’s locations in the liver. The pneumoperitoneum was generated with CO_2_ at a pressure of 12–15 mmHg, and visual exploration of the abdominal cavity was conducted with a 30° laparoscope. Intra-abdominal sonography of the liver was performed in order to plan the surgery. Adhesiolysis and liver mobilization were performed using the LigaSure™ device (LigaSure 5 mm; Valleylab, Boulder, CO, USA). Liver resections were carried out using LigaSure™, Endo GIA™ staplers (vascular cartridge, Endo GIA™, Covidien, Norwalk, CT, USA), and Cavitron Ultrasonic Surgical Aspirator (CUSA; Valleylab, Boulder, CO, USA). The specimen was then extracted through the hand-assisted device without a bag. An abdominal Jackson–Pratt (JP) drain was usually placed through a 5 mm port site. The port site’s fascia was closed in one layer of a continuous polydioxyanone (PDS) loop 1-0 suture (Ethicon, Johnson & Johnson, Bridgewater, NG, USA) and reinforced with interrupted Vicryl 2-0 sutures (Ethicon, Johnson & Johnson, Bridgewater, NG, USA). The 10 and 12 trocar sites were closed by Vicryl 1 (Ethicon, Johnson & Johnson, Bridgewater, NG, USA).

After discharge, the patients were followed by our multidisciplinary team during the first month, every four months in the first two years, and then twice a year with clinical examinations and blood work-up including carcinoembryonic antigen (CEA) and spiral CT of the chest–abdomen or PET-CT.

Blood loss was estimated using the volume of blood aspirated from the abdominal cavity during the procedure. Operative time was defined as time elapsed from skin incision until closure. Postoperative hospital stay was defined as the number of hospitalized days from the first day after operation until the day of discharge. From the pathological report, we collected data on the number of metastases and the largest tumor diameter for each patient. Complications were classified according to the Clavien–Dindo grading system [8].

From the blood test that had been performed one day before surgery, we recorded albumin level, and neutrophil, lymphocyte, and platelet count. Preoperative serum NLR and PLR were calculated as the absolute neutrophil or platelet count divided by the absolute lymphocyte, respectively. The prognostic nutritional index (PNI) was calculated as 10 × albumin level (g/dL) + 0.005 × total lymphocyte count (per mm^3^). The systemic immune-inflammation index (SII) was calculated as (neutrophil × platelet)/lymphocyte. We used cut-offs of 2.5, 200, 52, and 600 for NLR, PLR, PNI and SII respectively. 

### 2.1. Radiological Assessment of Incisional Hernia 

CT scans were retrospectively reviewed for all patients at least one year after surgery. A radiologist, blinded to the original radiological report and clinical results, reviewed all scans for abdominal wall herniation. This was correlated with documented clinical findings. Postoperative incisional hernia was radiologically defined as a disruption in the fascia at the site of the surgery (Figure 1). 

### 2.2. Statistical Analysis

Data collection and statistical analysis were performed using Microsoft Excel (Microsoft, Redmond, DC, USA) and SPSS 22.0 (IBM, Armonk, NY, USA), respectively. Quantitative and qualitative variables were described as median (range) and frequency (percentage), respectively. Comparisons between the two groups were analyzed with Pearson’s chi-squared test, and Fisher’s exact test for categorical variables. Univariate Cox regression analysis was applied to determine risk factors for developing an IH. Cox proportional hazard regression was also used to identify the influence of each risk factor and different research groups. A *p* value of <0.05 was considered to be statistically significant.

## 3. Results

Between January 2009 and December 2019, 126 patients underwent hand-assisted laparoscopic liver resection at the Department of Surgery, Carmel Medical Center, Israel. Of these patients, 89 (72.3%) met the inclusion criteria, and 31 patients were excluded from the study since they had different pathologies than metastatic CRC or because of insufficient data and follow-up (Figure 2). 

Our study population was divided into two subgroups depending on the radiological evidence of incisional hernia at one year follow-up. Incisional hernia was diagnosed in 7 (7.9%) patients; all of them were symptom-free and did not undergo surgical repair. Both groups shared nearly the same demographic characteristics without statistically significant differences, as shown in Table 1. All patients diagnosed with IH were obese according to the WHO classification, with a BMI of 25 or more (*p* = 0.03). The characteristics of the liver metastatic lesions (number and size of lesions) did not show any significant relationship with the occurrence of IH (*p* = 0.16 and 0.12, respectively). Neoadjuvant-chemotherapy and biological treatments (bevacizumab (Avastin^®^)) also did not show any significant difference between groups (*p* = 0.67 and 0.91, respectively). On the other hand, from operative factors, only simultaneous colon and liver resection was significantly different between groups, 2 (33.3%) patients with simultaneous resection had IH, compared to 5 (6%) patients in the liver-resection-only group (*p* = 0.016). The amount of intraoperative blood loss, the rate of blood transfusion, and complication rate were not different between groups (*p* = 0.97 and 0.79, respectively). 

Statistical relations between preoperative blood inflammatory markers and incidence of IH were investigated and analyzed as shown in Table 2. 

Albumin level, neutrophil-to-lymphocyte ratio (NLR > 2.5), prognostic nutritional index (PNI > 52), systemic immune-inflammation index (SII > 600), and platelet-to-lymphocyte ratio (PLR ≤ 200) were not significantly different between the two groups (*p* values = 0.46, 0.46, 0.27, and 0.09, respectively).

In multivariant analysis (Table 3), two independent risk factors for IH were identified: simultaneous liver and colon resection (HR 0.05, 95% CI 0.007–0.38, *p* = 0.004), and platelet-to-lymphocyte ratio (PLR ≤ 200) (HR 0.13, 95% CI 0.02–0.81, *p* = 0.02).

## 4. Discussion

IH remains a common complication in abdominal surgery, and its incidence plays an important role in the evaluation of the surgical technique. To the best of our knowledge, this is the largest observational study describing the incidence and risk factors of IH following HALS for CRLM.

The use of medical imaging has improved the ability to detect incisional hernias as compared to physical examination alone. Moreover, computed tomography (CT) is the gold standard to diagnose incisional hernia [9].

The rate of IH diagnosed by CT was 7.8% in our series. This favorably compares with the results of Wabitsch and his colleagues [10] in their newly published study. In their cohort of 18 patients who had undergone HALS for liver tumor, the incidence of IH was 22%. Moreover, our result compares well with an IH rate of 7.6% for the classical multi-incision laparoscopic liver surgery (MILS) approach in a cohort reported by Kazaryan and colleagues for colorectal liver metastasis after a median follow-up of 35.3 months [11]. In open HPB surgery, others reported an IH rate of 21.6% [12]. Our findings indicate that the HALS procedure for CRLM is not a risk factor for developing an IH when comparing it to the literature of open or pure laparoscopic approaches. 

According to many studies, obesity is considered to be a major risk factor in the development of incisional hernia, with 20% to 28% incidence of IH within 12 to 28 months after surgery [13]. In our study, obese patients had a higher rate of IH. Specifically, 100% of the patients who had developed IH had a BMI ≥ 25, compared to 0% of patients with BMI < 25. However, BMI ≥ 25 was not a risk factor for IH development in univariate analysis. 

The most common indication for liver resection is colorectal liver metastasis (CRLM). Prospective randomized and retrospective studies showed better survival after sequential modality treatment with neoadjuvant treatment, surgery, and adjuvant chemotherapy [9]. Neoadjuvant treatment includes chemotherapy regimens (5-fluorouracil (FU) and leucovorin with oxaliplatin (FOLFOX) or with irinotecan (FOLFIRI)) and biological agents such as vascular endothelial growth factor antibodies (VEGF) or antiepidermal growth factor receptor (EGFR). One of the side effects of the neoadjuvant treatment, especially anti-VEGF, is wound-healing complications [14]. According to the literature, preoperative chemotherapy was independently associated with an increased incidence of incisional hernia with a hazard ratio of 2.12 [15]. Bevacizumab (Avastin^®^) is a monoclonal anti-VEGF antibody that could impair wound healing and is associated with high incidence of wound complications, leading to more IH [16]. However, our results showed no significant association between Avastin^®^ and the rate of IH. This can be explained by the fact that we routinely performed surgery after at least 6 weeks interval from the last dose of the neoadjuvant treatment. 

In line with the results of Wabitsch and his colleagues, we also found that smoking status, diabetes, and the occurrence of postoperative complications were not associated with the development of an IH [10]. 

Systemic inflammatory responses play a crucial role in the pathogenesis and progression of cancer [17]. Inflammatory indicators, such as neutrophil-to-lymphocyte ratio (NLR), platelet-to-lymphocyte ratio (PLR), lymphocyte-to-monocyte ratio (LMR), prognostic nutritional index (PNI), and systemic immune-inflammation index (SII) were identified as prognostic indicators for disease-free and overall survival in resected CRLM [18]. Moreover, studies demonstrated their predictive role on both the progress and prognosis of many internal diseases, and infectious complications following various surgical procedures such as cesarean section, rectal cancer surgery, curative gastrectomy for gastric cancer, and hepatectomy. Recent studies also generalize the application of NLR to the field of hernia surgery, and it was significantly associated with persistent postsurgical complications in the presence of a strangulated inguinal hernia and with bowel resection in incarcerated groin hernia patients [19,20,21,22,23,24,25]. 

In this study, we investigated for the first time the relationship between those inflammatory markers and the incidence of IH after HALS for CRLM. Preoperative PLR ≤ 200 is an independent risk factor for the development of IH. Our results are in line with those of Maruyama and his colleges, who found in their study that PLR below 160 is significantly associated with the rate of wound-healing failure in patients after microsurgical head and neck reconstruction [26].

The secretion of *α* granules from activated platelets at the onset of injury is crucial for the initiation of wound healing, and of the coagulation and hemostasis, inflammatory, proliferative, and remodeling phases. α granules contain growth factors such as platelet- derived growth factor, transforming growth factor *β*, epidermal growth factor, vascular–endothelial growth factor; these factors are essential for the growth of local keratinocytes, fibroblasts, and accelerating angiogenesis [27].

Both platelets and lymphocytes are derived from the same hematopoietic stem cells, and the PLR should be kept constant for normal homeostasis. However, due to the different lifespans of platelets (10 days) and lymphocytes (several weeks to months) [28], in an abnormal state of hematopoiesis, the platelet count decreases faster than that of lymphocytes, leading to reduced PLR. In the present study, patients with a decreased PLR were in a transient state of peripheral blood abnormality; this may have been induced by decreased platelet production due to malignancy, myelosuppression by radiation therapy or chemotherapy [29], or decreased thrombopoietin production in the liver [30] or increased platelet consumption due to accelerated coagulation by tissue destruction or malignancy [31]. This systemic abnormality may lead to a decrease in the PLR and eventual reduction in the number of platelets that circulate and migrate to the microenvironment of the wound, causing direct impairment in the wound-healing process and in the long-term increase in IH incidence. 

Simultaneous liver and colon resection is an independent risk factor for IH. Snyder et al. showed in their analysis on the American College of Surgeons (ACS) NSQIP data that, although simultaneous resection offers definitive resection for patients with synchronous CRC and CRLM, it is associated with significantly increased 30-day overall morbidity and procedure-specific postoperative morbidity including anastomotic failure, bile leak, and hepatic dysfunction. Increased morbidity after simultaneous resection may be directly related to the incidence of IH [32]. 

The weaknesses of the study are mostly due to the retrospective design and the relatively few patients with risk factors. Prospective, larger, and randomized trials would define risk factors more for the development of incisional hernia after hand-assisted laparoscopic liver surgery for metastatic colorectal cancer. In addition, the clinical significance of the occurrence of incisional hernia was not deeply evaluated in our study. However, knowing the preoperative risk factors for IH (low PLR, simultaneous resection of liver and colon) can help in monitoring the patients with high risk for IH development and eventually consider postoperative precautions to reduce its incidence, such as using abdominal binders for a longer period after the surgery. In our study, correlation between PLR and IH after hand-assisted laparoscopic liver surgery for metastatic colorectal cancer was reported for the first time. Our results could lead to a prospective and randomized trial to evaluate if the correction and control of the PLR ratio before surgery changes the IH rate after hand-assisted laparoscopic liver surgery for metastatic colorectal cancer. 

## 5. Conclusions

The present study showed that HALS for CRLM had the same incidence of IH when compared with the pure laparoscopic liver resection literature. Independent risk factors for postoperative incisional hernia are simultaneous liver and colon resection, and platelet-to-lymphocyte ratio ≤ 200.

## Figures and Tables

**Figure 1 jpm-12-00492-f001:**
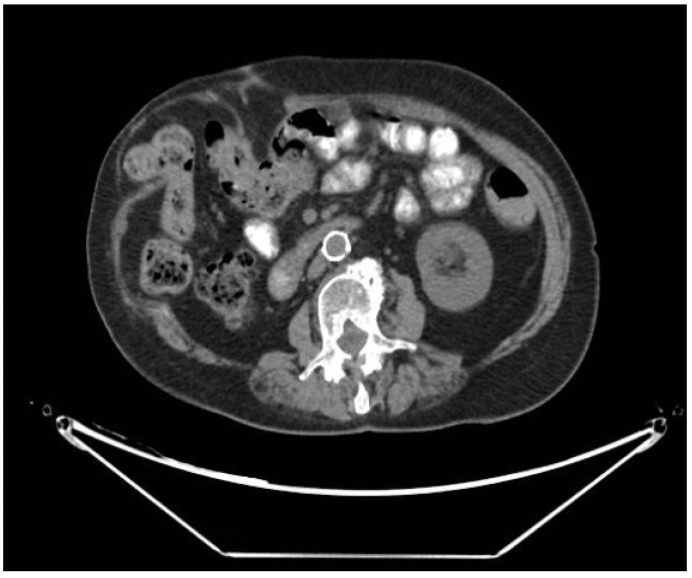
CT scan with incisional hernia in the right abdominal horizontal incision.

**Figure 2 jpm-12-00492-f002:**
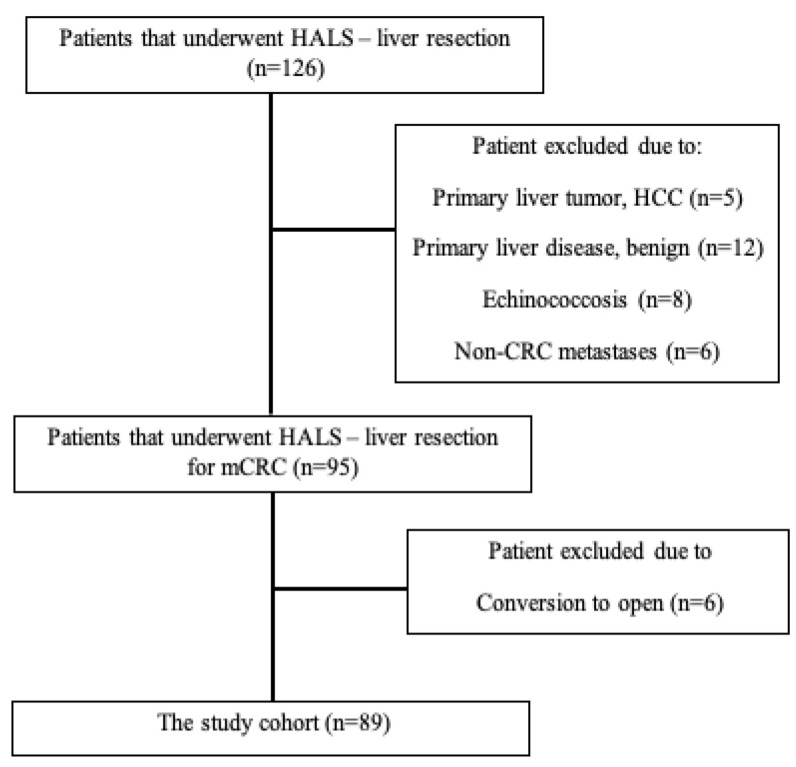
Prisma flowchart of the study.

**Table 1 jpm-12-00492-t001:** Patient demographic and clinical characteristics.

	All Cohort (*n* = 89)	Patients with Incisional Hernia (*n* = 7)	Patients without Incisional Hernia (*n* = 82)	*p*
Age	65.1 ± 12.2	67.6 ± 15.4	64.9 ± 11.9	0.59
Gender				0.46
Male	50 (56.2%)	3 (6%)	47 (94%)	
Female	39 (43.8%)	4 (10.3%)	35 (89.7%)	
Diabetes mellitus				0.86
Yes	23 (25.8%)	2 (8.7%)	21 (91.3%)	
No	66 (74.2%)	5 (7.6%)	61 (92.4%)	
Smoking				0.38
Yes	15 (16.8%)	2 (13.3%)	13 (86.7%)	
No	74 (83.2%)	5 (6.8%)	69 (93.750	
BMI				0.037
<25	33 (37.1%)	0 (0%)	33 (100%)	
>25	56 (62.9%)	7 (12.5%)	49 (87.5%)	
ASA				0.49
2-1	63 (70.8%)	6 (9.5%)	57 (90.5%)	
4-3	26 (29.2%)	1 (3.8%)	25 (96.2%)	
No. of liver metastases (mean ± SD)	1.76 ± 1.22	1.14 ± 0.38	1.81 ± 1.25	0.16
Size (mean ± SD)	31 ± 4	23.77 ± 10	32.14 ± 4	0.12
Neoadjuvant chemotherapy				0.67
Yes				
No	58 (65.2%)	4 (6.9%)	54 (93.1%)	
	31 (34.8%)	3 (9.7%)	28 (90.3%)	
Avastin				0.91
Yes	27 (30.3%)	2 (7.4%)	25 (92.6%)	
No	62 (69.7%)	5 (8.1%)	57 (91.9%)	
Simultaneous colon resection				0.016
Yes				
No	6 (6.7%)	2 (33.3%)	4 (66.7%)	
	83 (93.3%)	5 (6%)	78 (94%)	
EBL(mL) (mean ± SD)	337.2 ± 327.84	385.71 ± 484.83	332.9+324.81	0.68
Blood transfusion				0.97
Yes	13 (14.6%)	1 (7.7%)	12 (92.3%)	
No	76 85.4%)	6 (7.9%)	70 (92.1%)	
Postop complication				0.79
Yes	16 (18%)	1 (6.3%)	15 (93.8%)	
No	73 (82%)	6 (8.2%)	67 (91.8%)	

BMI, body mass index; ASA, American Society of Anesthesiologists; EBL, estimated blood loss.

**Table 2 jpm-12-00492-t002:** Preoperative blood test and nutritional indices.

	Entire Cohort (*n* = 89)	Patients with Incisional Hernia (*n* = 7)	Patients without Incisional Hernia (*n* = 82)	*p*
Albumin				0.45
<3.5	6 (6.7%)	0 (0%)	6 (100%)	
>3.5	83 (93.2%)	7 (8.4%)	76 (81.5%)	
NLR				0.46
<2.5	52 (58.4%)	5 (9.6%)	47 (90.4%)	
>2.5	37 (41.6%)	2 (5.4%)	35 (95.6%)	
PLR				0.09
<200	80 (89.9%)	5 (6.3%)	75 (93.8%)	
>200	9 (10.1%)	2 (22.2%)	7 (77.8%)	
PNI				0.46
<52	52 (58.4%)	5 (9.6%)	47 (90.4%)	
>52	37 (41.6%)	2 (5.4%)	35 (94.6%)	
SSI				0.27
<600	76 (85.4%)	5 (6.6%)	71 (93.4%)	
>600	13 (14.6%)	2 (15.4%)	11 (84.6%)	

NLR, neutrophil-to-lymphocyte ratio; PLR, platelet-to-lymphocyte ratio; PNI: prognostic nutritional index; SII, systemic immune-inflammation index.

**Table 3 jpm-12-00492-t003:** Univariate and multivariate cox regression analysis.

	Univariate		Multivariate	
	*p*	HR	*p*	HR
Diabetes mellitus No vs. yes	0.98	1.01 (0.18–5.59)		
Smoking No vs. yes	0.16	3.43 (0.61–19.31)		
BMI <25 vs. >25	0.15	0.011 (0.00–5.62)		
ASA 1–2 vs. 3–4	0.5	2.08 (0.24–17.97)		
Simultaneous colon No vs. yes	0.004	0.052 (0.007–0.38)	0.03	0.1 (0.01–0.83)
PLR <200 vs. >200	0.029	0.13 (0.02–0.81)	0.04	0.11(0.01–0.97)

BMI, body mass index; ASA, American Society of Anesthesiologists; PLR, platelet-to-lymphocyte ratio.

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
