# Peer review of "The Role of Preoperative Platelet-to-Lymphocyte Ratio as a Predictor for Incisional Hernias after Hand-Assisted Laparoscopic Liver Surgery for Metastatic Colorectal Cancer"

_jpm, 2022, doi:10.3390/jpm12030492_

Round 1
Reviewer 1 Report
Although an interesting paper tor read just a retrospective analysis of a Hand assisted experience. We already know that obesity , hand hamnd assisted and more than two surgical procedures done tigheter are risk for IH!
Author Response
We thank the editorial board and the reviewer for their careful review of our submitted manuscript. Please find below a point-by-point response addressing their comments
We agree with the reviewer on the fact that obesity, and two surgical procedures are known as risk factors for Incisional Hernia in conventional open surgical approach. In this study we demonstrated that this relation holds significantly not only in open surgical approach, but also in hand assisted approach. However, we found that hand assistant approach by itself is not a risk factor for incisional hernia development.
In discussion:
Line 178-186:
"The rate of IH, diagnosed by CT, found to be 7.8% at our series. this compares favorability with the results of Wabitsch and his colleagues (10) in their newly published study. In their cohort of 18 patients who underwent HALS for liver tumor, the incidence of IH was 22%. Moreover, our result compares well with IH rate of 7.6% for the classical multi-incision laparoscopic liver surgery (MILS) approach in a cohort reported by Kazaryan and colleagues for colorectal liver metastasis after a median follow-up of 35.3 months (11). In open HPB surgery, others have reported a rate of IH of 21.6% (12). Our findings indicate that the HALS procedure for CRLM is not a risk factor for developing an IH when comparing it to the literature of open or the pure laparoscopic approaches."
Line 187-191:
"According to many studies, obesity considered a major risk factor in the development of incisional hernia, with 20% to 28% incidence of IH within 12 to 28 months after surgery (13). In our study we observed that obese patients were noted to have a higher rate of IH. Specifically, 100% of the patients who have developed IH had a BMI ≥ 25 comparing to 0% of patients with BMI<25.
Line 248-254:
"We also found that simultaneous liver and colon resection is independent risk factor for IH. Snyder et al showed in their analysis on the American College of Surgeons (ACS) NSQIP data, that although simultaneous resection offers definitive resection for patients with synchronous CRC and CRLM, it is associated with significantly increased 30-day overall morbidity and procedure-specific postoperative morbidity including: anastomotic failure, bile leak, hepatic dysfunction etc. we believe that the increased morbidity after simultaneous resection is related directly to the incidence of IH (32). "
Reviewer 2 Report
The present paper is on the risk of developing incisional hernia in patients undergoing hand-assisted laparoscopic liver resection for metastasis from colorectal cancer. A total of 89 patients were included in this retrospective study covering an inclusion period of 10 years. Six of the patients underwent a simultaneous bowel resection.
All patients underwent follow-up with regular CT-scans and seven patients developed incisional hernia within the observation period. Multivariate analysis showed that a simultaneous bowel resection and the platelet-to-lymphocyte ratio £200 were risk factors for the development of incisional hernia. It is not possible to read from the text, whether all hernias were clinical or just confirmed at CT-scan. Did any of the patients undergo reoperation for the hernia?
It is difficult to read how the fascia at the hand-assisted port was closed. Was it one or two layers? Does Vicryl II mean “Vicryl 2-0” etc.?
Although the authors claim that this is the largest series reported on the hand-assisted operation this itself is not a quality measure. The figures are still very small with a high risk of type II error. What are the clinical consequences of the study, and how can the risk be reduced?
Platelet-to-lymphocyte ratio may be a marker for inflammation among several others and it is remarkable that the systemic imune-inflammation index was similar in patients with and without the development of hernia.
The manuscript could be shortened, and with a primary focus upon the development of incisional hernia and not so much on the liver resection.
Author Response
We thank the editorial board and the reviewer for their careful review of our submitted manuscript. Please find below a point-by-point response addressing their comments
We agree with the fact that we have relatively small cohort with high risk for type II error, being one of the limitations of our study. However, in the best of our knowledge it is the largest series to evaluate incisional hernia incidence after Hand-Assisted Laparoscopic Liver Surgery for Metastatic Colo-Rectal Cancer. We believe that knowing the PLR preoperatively can help in monitoring the patients with high risk for IH and eventually consider postoperative precautions to reduce the hernia incidence, like using abdominal binder for longer time after the surgery. We can also try to correct the PLR before surgery in order to reduce the postoperative complications and the occurrence of IH.
This clarification was added on the discussion section, line 257-267:
"The weaknesses of the study are mostly due to the retrospective design and the relatively few patients with risk factors. We believe that prospective, larger and randomized trials would define more the risk factors for the development of incisional hernia after Hand-Assisted Laparoscopic Liver Surgery for Metastatic Colo-Rectal Cancer. In addition, the clinical significance of the occurrence of incisional hernia was not deeply evaluated in our study. However, we believe that knowing the preoperative risk factors for IH (low PLR, simultaneous resection of liver and colon) can help in monitoring the patients with high risk for IH development and eventually consider postoperative precautions to reduce its incidence, like using abdominal binder for longer time after the surgery. Furthermore, we also can try to correct the PLR before surgery to reduce the risk for postoperative complications including IH development.
- Platelet-to-lymphocyte ratio may be a marker for inflammation among several others and it is remarkable that the systemic immune-inflammation index was similar in patients with and without the development of hernia.
The calculation of each marker is different than the other, and our goal was to check all the markers and try to find correlation between them and the outcomes. Change in the SII does not mean change in the PLR.
See lines 96-102, materials and methods section:
"From the blood test that performed one day before surgery we collect the Albumin level, neutrophil, lymphocyte and platelet count. Pre-operative serum NLR and PLR were calculated as the absolute neutrophil or platelet count divided by the absolute lymphocyte, respectively. The prognostic nutritional index (PNI) was calculated as 10 * albumin level (g/dl) + 0.005 * total lymphocyte count (per mm3). The systemic immune-inflammation index (SII) was calculated as (neutrophil * Platelet)/lymphocyte. We used the cut-off 2.5, 200, 52 and 600 for NLR, PLR, PNI and SII respectively."
The manuscript could be shortened, and with a primary focus upon the development of incisional hernia and not so much on the liver resection.
We tried to shorten it more, but the explanation about the hand assisted technique was crucial to explain the difference between it and the other approaches.
Reviewer 3 Report
This study adequatly shows risk factors for incisional hernia after hand assisted liver surgery. Is explains the risks wel, the design is appropriate. Conclusions are sound. Although is is a relatively large series the incisional hernia rate is low. Therfore this is a significant weakness in the study making th conclusions less firm. The use of English is exelent with only a few minor splling erors thata need correction.
Author Response
We thank the editorial board and the reviewers for their careful review of our submitted manuscript. Please find below a point-by-point response addressing their comments
We agree with the reviewer about the weakness of the study.
This clarification was added on the discussion section, line 257-267:
"The weaknesses of the study are mostly due to the retrospective design and the relatively few patients with risk factors. We believe that prospective, larger and randomized trials would define more the risk factors for the development of incisional hernia after Hand-Assisted Laparoscopic Liver Surgery for Metastatic Colo-Rectal Cancer. In addition, the clinical significance of the occurrence of incisional hernia was not deeply evaluated in our study. However, we believe that knowing the preoperative risk factors for IH (low PLR, simultaneous resection of liver and colon) can help in monitoring the patients with high risk for IH development and eventually consider postoperative precautions to reduce its incidence, like using abdominal binder for longer time after the surgery. Furthermore, we also can try to correct the PLR before surgery to reduce the risk for postoperative complications including IH development.
Round 2
Reviewer 1 Report
Although I do appreciate the effort done by the Authors to ameliorate their paper, really not a consistent change. Too many missing and biases
Author Response
We deeply thankful for the valuable comments of the reviewer.
We did change few paragraph in the discussion to underscore the weakness of our study including the biases- lines 256-259 "The weaknesses of the study are mostly due to the retrospective design and the relatively few patients with risk factors. We believe that prospective, larger and randomized trials would define more the risk factors for the development of incisional hernia after Hand-Assisted Laparoscopic Liver Surgery for Metastatic Colo-Rectal Cancer. In addition, the clinical significance of the occurrence of incisional hernia was not deeply evaluated in our study. However, we believe that knowing the preoperative risk factors for IH (low PLR, simultaneous resection of liver and colon) can help in monitoring the patients with high risk for IH development and eventually consider postoperative precautions to reduce its incidence, like using abdominal binder for longer time after the surgery. In our study we report for the first time the correlation between PLR and IH after Hand-Assisted Laparoscopic Liver Surgery for Metastatic Colo-Rectal Cancer, we hope that our results will lead to the conduction of a prospective and randomized trial to evaluate if the correction and control of the PLR ratio before the surgery will change the IH rate after Hand-Assisted Laparoscopic Liver Surgery for Metastatic Colo-Rectal Cancer."
In liver resection surgery, regardless of the minimal invasive technique, we need to make adequate incision to extract the specimen. This incision can be made as vertical midline incision or horizontal in the lower abdomen. In our study we used the same incision of the hand port to extract the specimen. During any surgery, fascial opening is prone for the development of hernia. We agree with the reviewer on the fact that obesity, and two surgical procedures are known as risk factors for Incisional Hernia in conventional open surgical approach. In this study we demonstrated that this relation holds significantly not only in open surgical approach, but also in hand assisted approach. However, we found that hand assistant approach by itself is not a risk factor for incisional hernia development. despite the up mentioned biases, we truly believe that our results are very important to evaluate the role of hand assisted surgery in the liver field. Especially that Our study is the biggest cohort that reports the incidence and risk factors for post-operative incisional hernia (IH) after for Hand-Assisted Laparoscopic Liver Surgery Colo-rectal liver metastasis.
In discussion:
Line 178-186:
"The rate of IH, diagnosed by CT, found to be 7.8% at our series. This compares favorability with the results of Wabitsch and his colleagues (10) in their newly published study. In their cohort of 18 patients who underwent HALS for liver tumor, the incidence of IH was 22%. Moreover, our result compares well with IH rate of 7.6% for the classical multi-incision laparoscopic liver surgery (MILS) approach in a cohort reported by Kazaryan and colleagues for colorectal liver metastasis after a median follow-up of 35.3 months (11). In open HPB surgery, others have reported a rate of IH of 21.6% (12). Our findings indicate that the HALS procedure for CRLM is not a risk factor for developing an IH when comparing it to the literature of open or the pure laparoscopic approaches."
Line 187-191:
"According to many studies, obesity considered a major risk factor in the development of incisional hernia, with 20% to 28% incidence of IH within 12 to 28 months after surgery (13). In our study we observed that obese patients were noted to have a higher rate of IH. Specifically, 100% of the patients who have developed IH had a BMI ≥ 25 comparing to 0% of patients with BMI<25.
Line 248-254:
"We also found that simultaneous liver and colon resection is independent risk factor for IH. Snyder et al showed in their analysis on the American College of Surgeons (ACS) NSQIP data, that although simultaneous resection offers definitive resection for patients with synchronous CRC and CRLM, it is associated with significantly increased 30-day overall morbidity and procedure-specific postoperative morbidity including: anastomotic failure, bile leak, hepatic dysfunction etc. we believe that the increased morbidity after simultaneous resection is related directly to the incidence of IH (32). "
Reviewer 2 Report
Thank your for your comments and revison. Still have some minor comments.
Line 80: what is a "loop I suture"? Do you mean a 1-0 suture?
Line 81: Why was the fascia at the port sites reinforced with Vicryl 2-0?
Line 266: Do you really think that correction of the PLR will prevent the development of incisional hernia. The statistical finding of the PLR as a predictor for the development of hernia could be the results of a type 2 error!?
Line 263 to 266 should be omitted as it is very speculative and imprecise.
Author Response
We deeply thankful for the valuable comments of the reviewer.
Thank you for your comments and revision. Still have some minor comments.
- Line 80: what is a "loop I suture"? Do you mean a 1-0 suture?
Response:
Yes, and we clarified that in the manuscript, line 81 " loop 1-0 suture"
- Line 81: Why was the fascia at the port sites reinforced with Vicryl 2-0?
Response:
Our practice is to use interrupted Vicryl 2-0 suture to reinforce the continuous PDS (Polydioxyanone ) loop 1-0 suture . Especially in the edges of the incision.
- Line 266: Do you really think that correction of the PLR will prevent the development of incisional hernia. The statistical finding of the PLR as a predictor for the development of hernia could be the results of a type 2 error!?
Response:
We agree with the fact that PLR as predictor for HI could be the result of type 2 error. And as we mentions in the discussion (lines 257-259) a prospective, larger and randomized trials would define more the risk factors for the development of incisional hernia after Hand-Assisted Laparoscopic Liver Surgery for Metastatic Colo-Rectal Cancer. And then the next sept would be – prospective and randomized trial to evaluate if the correction and control of the PLR ratio before the surgery will change the IH rate after Hand-Assisted Laparoscopic Liver Surgery for Metastatic Colo-Rectal Cancer.
This clarification and changes were added on the discussion section, lines 264-269, " In our study we report for the first time the correlation between PLR and IH after Hand-Assisted Laparoscopic Liver Surgery for Metastatic Colo-Rectal Cancer, we hope that our results will lead to the conduction of a prospective and randomized trial to evaluate if the correction and control of the PLR ratio before the surgery will change the IH rate after Hand-Assisted Laparoscopic Liver Surgery for Metastatic Colo-Rectal Cancer. "
- Line 263 to 266 should be omitted as it is very speculative and imprecise.
Response:
we agree with the valuable comment. This clarifications and changes were added on the discussion section, lines 264-269, " In our study we report for the first time the correlation between PLR and IH after Hand-Assisted Laparoscopic Liver Surgery for Metastatic Colo-Rectal Cancer, we hope that our results will lead to the conduction of a prospective and randomized trial to evaluate if the correction and control of the PLR ratio before the surgery will change the IH rate after Hand-Assisted Laparoscopic Liver Surgery for Metastatic Colo-Rectal Cancer. "